# The Response Regulator OmpR Negatively Controls the Expression of Genes Implicated in Tilimycin and Tilivalline Cytotoxin Production in *Klebsiella oxytoca*

**DOI:** 10.3390/microorganisms13010158

**Published:** 2025-01-14

**Authors:** Ramón G. Varela-Nájera, Miguel A. De la Cruz, Jorge Soria-Bustos, Carmen González-Horta, Ma Carmen E. Delgado-Gardea, Jorge A. Yáñez-Santos, María L. Cedillo, Hidetada Hirakawa, James G. Fox, Blanca Sánchez-Ramírez, Miguel A. Ares

**Affiliations:** 1Unidad de Investigación Médica en Enfermedades Infecciosas y Parasitarias, Hospital de Pediatría, Centro Médico Nacional Siglo XXI, Instituto Mexicano del Seguro Social, Mexico City 06720, Mexico; p299553@uach.mx; 2Facultad de Ciencias Químicas, Universidad Autónoma de Chihuahua, Circuito Universitario Campus II, Chihuahua 31125, Mexico; carmengonzalez@uach.mx (C.G.-H.); mcdelgado@uach.mx (M.C.E.D.-G.); bsanche@uach.mx (B.S.-R.); 3Centro de detección Biomolecular, Benemérita Universidad Autónoma de Puebla, Puebla 72592, Mexico; miguel.delacruzv@correo.buap.mx (M.A.D.l.C.); jorge_s41@hotmail.com (J.S.-B.); jorge.yanez@correo.buap.mx (J.A.Y.-S.); lilia.cedillo@correo.buap.mx (M.L.C.); 4Facultad de Medicina, Benemérita Universidad Autónoma de Puebla, Puebla 72410, Mexico; 5Department of Bacteriology, Graduate School of Medicine, Gunma University, Maebashi 371-8514, Japan; hirakawa@gunma-u.ac.jp; 6Division of Comparative Medicine, Massachusetts Institute of Technology, Cambridge, MA 02139, USA; jgfox@mit.edu; 7Departamento de Microbiología, Escuela Nacional de Ciencias Biológicas, Instituto Politécnico Nacional, Mexico City 11340, Mexico

**Keywords:** *Klebsiella oxytoca*, OmpR, *aroX*, *npsA*, tilimycin, tilivalline

## Abstract

*Klebsiella oxytoca* toxigenic strains represent a critical health threat, mainly due to their link to antibiotic-associated hemorrhagic colitis. This serious condition results from the bacteria’s ability to produce tilimycin and tilivalline cytotoxins. Our research highlights the pivotal role of OmpR, a key regulator within the EnvZ/OmpR two-component system, in controlling the virulence factors associated with *K. oxytoca*. Our findings strongly indicate that OmpR is a repressor of the *aroX* and *npsA* genes, the first genes of *aroX* and NRPS operons, respectively, which are indispensable for producing these enterotoxins. Notably, in the absence of OmpR, we observe a significant increase in cytotoxic effects on Caco-2 cells. These observations identify OmpR as a crucial negative transcription regulator for both operons, effectively managing the release of these cytotoxins. This research deepens our understanding of the mechanisms of toxigenic *K. oxytoca* and opens promising avenues for targeting OmpR for new therapeutic interventions. By focusing on this innovative approach, we can develop more effective solutions to combat this pressing health challenge, ultimately improving patient outcomes against this pathogen.

## 1. Introduction

*Klebsiella oxytoca*, a Gram-negative bacillus present in our intestinal microbiota, emerges as a notable pathogen partly due to its penicillin resistance. This resistance is linked to its intrinsic antimicrobial resistance genes, leading to the production of beta-lactamases [1]. When penicillin is used, a notable disruption occurs, where the delicate balance of gut microbiota is thrown off, often resulting in a state known as dysbiosis. Dysbiosis is marked by a decrease in beneficial microorganisms and the rise of potentially harmful ones, such as toxigenic *K. oxytoca*. This disruption is not just a minor issue; it significantly impacts gut health and stability, raising serious concerns that deserve our attention [2,3].

As toxigenic *K. oxytoca* proliferates in the large intestine, there is a notable increase in the production of tilimycin (TM) and tilivalline (TV) cytotoxins. These toxins can lead to antibiotic-associated hemorrhagic colitis (AAHC), presenting troubling symptoms such as bloody diarrhea, abdominal cramps, and rectal sparing [4,5,6].

The OmpR protein is a key player in the EnvZ/OmpR two-component system, which is essential in regulating virulence factors in pathogenic bacteria. This system is remarkable for its ability to sense environmental signals, such as changes in osmolarity and pH [7]. The EnvZ transmembrane histidine kinase plays a critical role by detecting these changes and phosphorylating the OmpR response regulator. In turn, OmpR binds to specific regions in DNA, fine-tuning gene expression as needed. Its pivotal involvement in this multifaceted regulatory system highlights its significance in the pathogenicity of bacteria [8].

Our research brings to light the crucial role of OmpR in regulating the expression of the *aroX* and *npsA* genes in the toxigenic strain *K. oxytoca* MIT 09-7231. We demonstrated that OmpR serves as a key repressor for both genes, playing an essential role in the pathogenic behavior of this bacterium. Additionally, our study on the cytotoxic effects of *K. oxytoca* on Caco-2 cells reveals the profound influence of OmpR and indicates that the absence of OmpR resulted in a significant increase in cytotoxic response, whereas its overexpression markedly diminished this response. These compelling findings underscore the function of OmpR in repressing genes linked to TM/TV biosynthesis, highlighting its critical importance in the pathogenicity of toxigenic *K. oxytoca*. The implications of our research are promising and could significantly enhance our understanding of gut health and interactions with pathogens.

## 2. Materials and Methods

We used the *Klebsiella oxytoca* strain MIT 09-7231, a well-characterized and prototypic toxigenic strain [9]. The strains of *K. oxytoca* and the plasmids utilized in this study are listed in Table 1. A key aspect of our strategy involved the lambda Red recombinase system [10], which allowed us to create the Δ*ompR* mutant strain.

To accomplish our goal, we designed a PCR fragment featuring a kanamycin resistance cassette flanked by 45 bp on both the 5′ and 3′ ends, precisely outlining the boundaries of the *ompR* gene. This fragment was synthesized using the pKD4 plasmid as a template, incorporating specific primers (Appendix A). Following the purification of the PCR product with the QIAquick PCR Purification Kit (Qiagen, Hilden, Germany), we expertly electroporated it into competent *K. oxytoca* cells containing the pKD119 plasmid which expresses the lambda Red recombinase. We employed the MicroPulser Electroporator (Bio-Rad, Hercules, CA, USA) in accordance with the high-efficiency transformation protocol tailored for *Escherichia coli* (https://www.bio-rad.com/sites/default/files/2022-01/10000148532.pdf) accessed on 18 April 2023. Furthermore, we enhanced the expression of the lambda Red recombinase by adding L-(+)-arabinose at a final concentration of 1% to the culture medium. PCR and unequivocal sequencing confirmed the success of our mutation strategy [10].

We generated the Δ*ompR* p-OmpR-complemented strain by introducing the p-OmpR plasmid into the Δ*ompR* strain through electroporation using a MicroPulser Electroporator (Bio-Rad). The p-OmpR plasmid was created by subcloning the *ompR* gene from the pTRC99A-ompR plasmid [11] into the pTRC99K vector [12]. The process involved digesting the pTRC99A-ompR vector with *Nco*I and *Bam*HI restriction enzymes. The resulting *ompR* gene fragment was purified using the QIAquick Gel Extraction Kit (Qiagen) and ligated with T4 DNA ligase (Invitrogen, Waltham, MA, USA) into the pre-digested pTRC99K vector. DNA sequencing confirmed the identity. We also introduced the plasmid into the wild-type (WT) strain via electroporation to overexpress OmpR, a crucial step in our research. We added 50 µM isopropyl β-D-1-thiogalactopyranoside (IPTG) to the culture medium to facilitate this.

The strains were cultivated in tryptone soy broth (TSB) (Difco, Beirut, Lebanon) at 37 °C with shaking until they reached an optical density at 600 nm (OD) of 1.6 since, in a previous study, we demonstrated that the expression levels of the *aroX* and NRPS operons are significantly higher during the stationary growth phase [13]. By capturing gene expression at its peak, our approach ensures a more accurate and comprehensive understanding of the underlying biological processes.

Total RNA was isolated using the Quick-RNA Fungal/Bacterial Miniprep Kit (Zymo Research, Irvine, CA, USA) and then purified with the TURBO DNA-free Kit (Invitrogen) to remove any residual DNA. The concentration and purity of the RNA were assessed using a NanoDrop One (Thermo Scientific, Waltham, MA, USA), while its integrity was evaluated through a bleach-denaturing 1.5% agarose gel [14]. Reverse transcription (RT) was conducted using 1 µg of RNA with the Revertaid First Strand cDNA Synthesis Kit (Thermo Scientific). Control reactions that excluded the reverse transcriptase enzyme were included in each RT experiment.

Quantitative PCR (qPCR) was performed using a LightCycler 480 instrument (Roche, Basel, Switzerland) with specific primers [13] (Appendix A) and the SYBR-Green detection method. To achieve optimal results, we prepared a reaction mix consisting of 2.0 μL of PCR-grade water, 0.5 μL (10 μM) of the forward primer, 0.5 μL (10 μM) of the reverse primer, 5 μL of 2X SYBR Green I Master (Roche), and 2.5 μL of cDNA (~25 ng). All samples were loaded into a multiwell plate for amplification in the LightCycler 480. Each sample was amplified in triplicate across three independent experiments. The *rrsH* gene (16S rRNA) was used as a reference for normalization to further validate our findings.

The optimized qPCR analysis included the following steps: (1) an initial denaturation at 95 °C for 10 min, followed by 45 amplification cycles lasting 10 s at 95 °C, 10 s at 59 °C, and 10 s at 72 °C, with a single fluorescence measurement taken during each cycle; (2) a melting curve program lasting 10 s at 95 °C and 1 min at 65 °C with continuous fluorescence measurement until reaching 97 °C; and (3) a final cooling period of 10 s at 40 °C. We rigorously confirmed the absence of contaminating DNA by demonstrating the lack of amplification products after 45 qPCR cycles when using RNA as a template. Moreover, the minus RT controls were included in all experiments. We calculated the fold change in gene expression using the 2^−ΔΔCt^ method [15], ensuring the accuracy of our findings.

The cytotoxicity was evaluated using the LDH Cytotoxicity Assay Kit (Invitrogen). This method measures the release of lactate dehydrogenase (LDH) from epithelial cells, in accordance with the manufacturer’s guidelines (https://assets.thermofisher.com/TFS-Assets%2FLSG%2Fmanuals%2FMAN0018500_CyQUANT-LDH-Cytotoxicity-Assay-Kit_PI.pdf) accessed on 26 September 2024. Our procedure involved inoculating 10 µL of filtered supernatants from the studied strains—WT, ∆*ompR*, ∆*ompR* p-OmpR, and WT p-OmpR (the latter cultured with 50 µM IPTG)—into 90 µL of Caco-2 cells. This resulted in a total of 1 × 10^4^ cells, which had been previously counted using the trypan blue exclusion method in a Neubauer chamber [16].

The Caco-2 cells were carefully cultured in Dulbecco’s Modified Eagle Medium (DMEM) with high glucose (4.5 g/L) (Gibco, Waltham, MA, USA), supplemented with 10% fetal bovine serum (FBS) (Gibco), in a 96-well flat-bottom culture plate. After incubating for 48 h at 37 °C in a 5% CO_2_ atmosphere—conditions established in previous studies as optimal for observing significant cytotoxic effects [3,9,13,17,18]—we transferred 50 µL of each sample to a new 96-well culture plate. We added the appropriate solutions from the kit. To assess cytotoxicity, we measured lactate dehydrogenase (LDH) levels by calculating the difference between the absorbance at 680 nm and that at 490 nm using a spectrophotometer (Multiskan Ascent, Thermo Scientific).

Negative controls included phosphate-buffered saline (PBS) and culture medium (TSB), while lysis buffer served as a positive control to validate our findings. Additionally, we analyzed the *K. oxytoca* Δ*npsA* mutant strain to confirm that the observed cytotoxicity was due to the production of TM/TV, as this strain lacks the ability to synthesize these cytotoxins [19,20].

All experiments were conducted in triplicates across three independent biological replicates. To ensure a thorough and comprehensive analysis, we performed an unpaired two-tailed Student’s *t*-test using GraphPad Prism 10.4 software (GraphPad Inc., San Diego, CA, USA). Values of *p* < 0.05 were considered statistically significant.

## 3. Results

### 3.1. OmpR Is a Repressor of aroX and npsA Gene Expression

We observed a significant up-regulation of the *aroX* and *npsA* genes, which are the initial genes in their respective operons (*aroX* and NRPS), in the Δ*ompR* strain. The expression of the *aroX* gene saw an impressive 6.8-fold increase, while the *npsA* gene exhibited a remarkable 9.5-fold rise compared to the wild-type (WT) strain. Even more compelling, the transcription levels in the complemented strain returned to those of the WT, highlighting the crucial role of OmpR in regulating the *aroX* and NRPS operons. This groundbreaking discovery opens the door to deeper insights into gene regulation and its potential implications for future research (Figure 1A,B).

We also observed a notable down-regulation of gene expression when OmpR was induced from the p-OmpR plasmid using IPTG in the WT strain. A striking 9.0-fold decrease in *aroX* gene expression and an outstanding 10.0-fold decrease in *npsA* gene expression compared to the non-induced strain were recorded. These results highlight the significant impact of OmpR on gene regulation (Figure 1C,D).

Combined with the restored gene expression in the complemented strain to WT levels, these results convincingly demonstrate that the OmpR protein functions effectively as a transcriptional repressor of the *aroX* and NRPS operons.

### 3.2. The Cytotoxic Effect on Colonic Epithelial Cells Is Affected by OmpR

The supernatant from the WT strain has clearly demonstrated significant cytotoxicity in Caco-2 cells, underscoring the serious threat that toxigenic *Klebsiella oxytoca* strains pose to epithelial cells. Remarkably, the supernatant from the mutant Δ*ompR* strain exhibited even greater levels of cytotoxicity, highlighting a dramatic increase in its pathogenic potential. Importantly, the supernatant from the complemented Δ*ompR* p-OmpR strain restored cytotoxicity to levels akin to the WT strain, reinforcing the validity of our experimental methodologies and results. Moreover, we observed that culturing the supernatant from the WT p-OmpR strain with IPTG to induce OmpR overexpression led to a striking reduction in cytotoxicity against Caco-2 cells (Figure 2).

To establish a direct link between the observed cytotoxic effects and the production of TM and TV compounds, we conducted experiments using the Δ*npsA* mutant strain, which lacks the ability to produce these compounds. As anticipated, the supernatant from the Δ*npsA* mutant did not induce cytotoxicity in Caco-2 cells. This finding confirms the critical roles that TM and TV compounds play in the cytotoxicity of toxigenic *K. oxytoca* against epithelial cells (Figure 2). This compelling evidence highlights the significant role of OmpR and its promising implications for the development of targeted therapeutic strategies.

## 4. Discussion

Understanding gene expression in bacteria is incredibly exciting due to their adaptability. By exploring gene regulation, we can unlock possibilities for personalized treatments, particularly for conditions like AAHC. This not only highlights bacteria’s resilience against infections but also our potential to combat them effectively. With advancements in gene regulation, particularly focusing on transcription factors, we are poised to revolutionize treatment approaches and pave the way for a healthier future.

This research powerfully highlights the pivotal role of OmpR in regulating gene expression in toxigenic *K. oxytoca*. As a key master regulator, OmpR is instrumental in activating and repressing gene transcription. Our comprehensive investigation focuses on how OmpR regulates *aroX* and *npsA*, the first genes in their respective *aroX* and NRPS operons. These operons are vital, as they encode essential enzymes that facilitate the biosynthesis of TM and TV cytotoxins. This emphasizes the significance of our findings and their implications for understanding toxin production in this microorganism [20,21,22,23,24].

Our findings indicate that OmpR effectively represses the transcription of the *aroX* and *npsA* genes. This repression aligns with previously documented regulatory effects of OmpR on various genes, including those in the *atp* and *gcvTHP* operons in *K. pneumoniae* [18]. Additionally, in *E. coli* K-12, OmpR negatively regulates *ompF*, *cadA*, *cadB*, and *cadC* [25], while in *Salmonella* Typhimurium, it influences *rpoS* repression [26]. In *Yersinia enterocolitica*, OmpR has regulatory effects on the transcriptional repression of *fur*, *yopD*, *yadA*, and *hemR* [27,28,29]. In *Vibrio cholerae*, OmpR represses *aph*, *oscR*, and *ompW* [30,31] and in *Pseudomonas syringae*, it negatively regulates *hrpR* and *hrpS* [32].

The TM and TV enterotoxins serve as the primary virulence factors in toxigenic strains of *K. oxytoca*, resulting in severe cytotoxic damage to epithelial cells. Our research reveals that the absence of the OmpR regulatory protein significantly enhances the transcription of the *aroX* and *npsA* genes, which in turn boosts the production of TM and TV toxins. This surge in toxin production leads to a marked increase in the cytotoxicity of the toxigenic *K. oxytoca* strain (MIT 09-7231) against Caco-2 cells. In this context, OmpR plays a crucial role as both a positive and negative transcriptional regulator. It not only amplifies the expression of the *stx-1* gene, responsible for encoding Shiga-like toxin 1 in enterohemorrhagic *E. coli*, but also effectively downregulates the ToxR regulon in *V. cholerae* [31,33]. This exceptional dual action results in a significant rise in Shiga-like toxin production while decreasing cholera toxin levels.

Overall, our findings demonstrate the significant role of the OmpR protein in negatively regulating the expression of the *aroX* and NRPS biosynthetic operons, thereby directly controlling the production of TM and TV compounds. This research not only enhances our understanding of the molecular mechanisms underlying toxin production in *K. oxytoca* but also opens the door to new insights into bacterial virulence.

As we progress with this innovative research project, our team is excited to explore whether OmpR directly binds to the intergenic regulatory region of *aroX* and *npsA*, thereby influencing the direct regulation of the *aroX* and NRPS operons. We plan to accomplish this by performing mobility shift assays. This investigation could significantly enhance our understanding of how OmpR regulates transcription and may have important implications for developing strategies to combat bacterial virulence factors. Gaining a deep understanding of OmpR’s complex functions is essential for appreciating its impact on microbial virulence, ultimately highlighting the importance of focusing research efforts on this regulatory protein.

## 5. Conclusions

This research emphasizes the crucial role of the OmpR protein in negatively regulating the *aroX* and NRPS biosynthetic operons. Understanding this regulatory mechanism provides valuable insights into how the toxigenic bacterium *K. oxytoca* produces TM and TV toxic compounds, enhancing our knowledge of the molecular processes involved in toxin production. Additionally, clarifying the importance of OmpR in the virulence of toxigenic *K. oxytoca* opens up exciting opportunities for future research, potentially leading to innovative strategies for addressing toxin-related challenges.

## Figures and Tables

**Figure 1 microorganisms-13-00158-f001:**
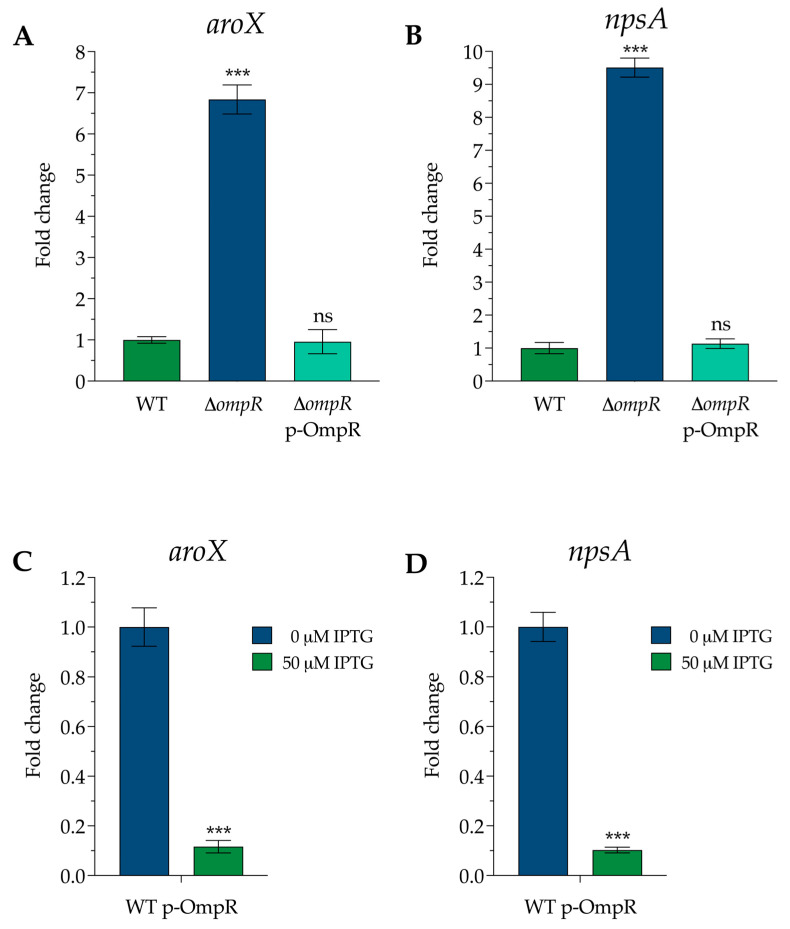
The effect of the OmpR protein in the regulation of *aroX* and *npsA* gene expression. This figure illustrates the transcriptional expression of (**A**) *aroX* and (**B**) *npsA* in three strains: wild-type (WT), mutant (Δ*ompR*), and complemented (Δ*ompR*-p-OmpR). Additionally, it shows the transcription levels of (**C**) *aroX* and (**D**) *npsA* in the WT p-OmpR strain after the addition of 50 µM IPTG. The data represent the means from three independent experiments conducted in triplicate, along with standard deviations. Statistically significant results are indicated as follows: *** *p* < 0.001; ns: not significant. All *p*-values were calculated using an unpaired two-tailed Student *t*-test.

**Figure 2 microorganisms-13-00158-f002:**
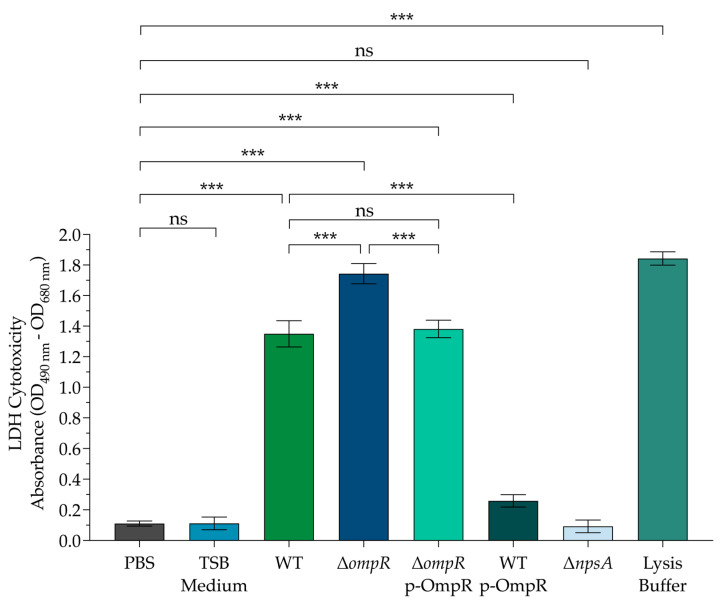
Effect of OmpR in the cytotoxicity of toxigenic *K. oxytoca* on colonic epithelial cells. Cultures of Caco-2 cells were inoculated with TSB medium, and toxigenic *K. oxytoca* supernatants WT, Δ*ompR*, Δ*ompR*-pOmpR, WT p-OmpR, and Δ*npsA* were incubated for 48 h. Afterwards, aliquots were used to measure the LDH release from Caco-2 cells. We determined the minimal and maximal LDH release by inoculating PBS and lysis buffer, respectively. The data represent the means from three independent experiments conducted in triplicate, along with standard deviations. Statistically significant results are indicated as follows: *** *p* < 0.001; ns: not significant. All *p*-values were calculated using an unpaired two-tailed Student *t*-test.

**Table 1 microorganisms-13-00158-t001:** Bacterial strains and plasmids used in this study.

Strain or Plasmid	Description *	Reference
Strains		
*K. oxytoca* WT	Wild-type *K. oxytoca* strain MIT 09-7231	[9]
*K. oxytoca* Δ*ompR*	*K. oxytoca* Δ*ompR::kan*	This study
*K. oxytoca* Δ*ompR* p-OmpR	*K. oxytoca* Δ*ompR::kan* + p-OmpR	This study
*K. oxytoca* WT p-OmpR	*K. oxytoca* WT + p-OmpR	This study
Plasmids		
pTrc99A-ompR	*ompR* expression plasmid, IPTG-inducible *trc* promoter, Amp^R^	[11]
pTrc99K	Expression plasmid, IPTG-inducible *trc* promoter, Kan^R^	[12]
p-OmpR	*ompR* expression plasmid, IPTG-inducible *trc* promoter, Kan^R^	This study
pKD119	pINT-ts derivative containing the lambda Red recombinase system under an arabinose-inducible promoter, Tet^R^	[10]
pKD4	pANTsy derivative template plasmid containing the kanamycin cassette for lambda Red recombination, Amp^R^	[10]

* Amp^R^, ampicillin resistance; Kan^R^, kanamycin resistance; Tet^R^, tetracycline resistance.

## Data Availability

The original contributions presented in the study are included in the article/Appendix A, further inquiries can be directed to the corresponding authors.

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
