# Peer review of "The Response Regulator OmpR Negatively Controls the Expression of Genes Implicated in Tilimycin and Tilivalline Cytotoxin Production in Klebsiella oxytoca"

_microorganisms, 2025, doi:10.3390/microorganisms13010158_

Round 1
Reviewer 1 Report
Comments and Suggestions for Authors
This paper confirmed the role of the OmpR protein in negatively regulating the expression of the aroX and NRPS biosynthetic operons, and affecting the production of TM and TV compounds. This paper is well-written, with clear logic and a well-organized structure. With appropriate revisions, the paper can be accepted.
1. Some critical experimental details are missing, particularly regarding the lambda-red recombinase system, the electroporation method, and the construction of the plasmids etc.
2. The practical implications of this research should be delved into more deeply.
Author Response
This paper confirmed the role of the OmpR protein in negatively regulating the expression of the aroX and NRPS biosynthetic operons, and affecting the production of TM and TV compounds. This paper is well-written, with clear logic and a well-organized structure. With appropriate revisions, the paper can be accepted.
Response: We sincerely appreciate the time you dedicated to reviewing our manuscript. Your insightful observations will greatly enhance the quality of our paper, and we are truly grateful for your feedback. To make it easier for you to see our revisions, we’ve highlighted our responses to your comments in yellow throughout the manuscript.
- Some critical experimental details are missing, particularly regarding the lambda-red recombinase system, the electroporation method, and the construction of the plasmids etc.
Response: In this revised version, the materials and methods section of the manuscript comprehensively addresses the previously missing experimental details related to the lambda-Red recombinase system, the electroporation technique, and the plasmid construction process, among other critical elements. We sincerely appreciate your insightful observation. lines 77-98.
2. The practical implications of this research should be delved into more deeply.
Response: In this updated version of the manuscript, we have highlighted the important practical implications of studying gene expression regulation in general (lines 219-224). Moreover, we have incorporated a dedicated paragraph that outlines the specific practical significance of our research findings (lines 259-264). We appreciate your insightful observations.
Reviewer 2 Report
Comments and Suggestions for Authors
I think the author's findings are quite interesting, OmpR as a key negative regulatory factor for the transcription of aroX and NRPS operons. Although the article is brief, it has certain guiding significance in controlling the production of these potent cell toxins. There are some minor issues that need to be resolved.
1) . The cytotoxic effect on colonic epithelial cells is affected by OmpR, Whether OMPR directly interacts with regulatory factors of aroX and NRPS or is mediated through an intermediate regulatory gene, it would be even more perfect if the author explains how to eliminate this.
2) . Table 1 can be attached as an attachment.
Author Response
I think the author's findings are quite interesting, OmpR as a key negative regulatory factor for the transcription of aroX and NRPS operons. Although the article is brief, it has certain guiding significance in controlling the production of these potent cell toxins. There are some minor issues that need to be resolved.
Response: We genuinely appreciate the time you invested in reviewing our manuscript. Your insightful observations have greatly enhanced the quality of our work. We've highlighted our responses to your comments in green throughout the manuscript for your convenience. Thank you for your support!
1) . The cytotoxic effect on colonic epithelial cells is affected by OmpR, Whether OMPR directly interacts with regulatory factors of aroX and NRPS or is mediated through an intermediate regulatory gene, it would be even more perfect if the author explains how to eliminate this.
Response: At this stage of our research, we cannot definitively conclude whether OmpR directly interacts with the regulatory regions of the aroX and NRPS operons or if this interaction is facilitated by an intermediate regulatory gene. Nevertheless, our team is enthusiastic about the prospect of advancing our innovative project. We are committed to investigating whether OmpR directly binds to the intergenic regulatory region of aroX and npsA. This crucial discovery could illuminate how OmpR influences the regulation of aroX and NRPS operons. To support this investigation, we will conduct mobility shift assays (lines 256-259).
2) . Table 1 can be attached as an attachment.
Response: Thank you for the suggestion. In this new version, we have included the previous Table 1 as Supplementary Table 1.
Reviewer 3 Report
Comments and Suggestions for Authors
The manuscript by Ramón G. Varela-Nájera and co-authors entitled "The response regulator OmpR negatively controls the expression of genes implicated in tilimycin and tilivalline cytotoxins production in Klebsiella oxytoca" is devoted to studying the influence of the regulator OmpR on the expression of the aroX and npsA genes. The manuscript is well-written, and I don't see any significant issues.
However, there are a few minor corrections needed:
- Lines 79-80: Italicize NcoI and BamHI.
- Line 96: What probes were used?
- There is no reference for source [14]; it jumps from [13] to [15].
- Table 1: Correct 'pb' to 'bp'
- Line 243: Why are the authors' names missing?
Author Response
The manuscript by Ramón G. Varela-Nájera and co-authors entitled "The response regulator OmpR negatively controls the expression of genes implicated in tilimycin and tilivalline cytotoxins production in Klebsiella oxytoca" is devoted to studying the influence of the regulator OmpR on the expression of the aroX and npsA genes. The manuscript is well-written, and I don't see any significant issues.
Response: We sincerely appreciate your effort in reviewing our manuscript. Your insightful observations have greatly contributed to its improvement. To help you see our responses, we have highlighted them in turquoise throughout the text. Thank you for your valuable feedback.
However, there are a few minor corrections needed:
- Lines 79-80: Italicize NcoI and BamHI.
Response: We have now italicized the first three letters related to the bacterial genus for the mentioned restriction enzymes. Thank you for pointing this out.
- Line 96: What probes were used?
Response: We used SYBR Green as the detection method along with specific primers to determine gene expression through qPCR. This information is now included in Supplementary Table 1.
- There is no reference for source [14]; it jumps from [13] to [15].
Response: This has been corrected.
- Table 1: Correct 'pb' to 'bp'
Response: In this new version, we have not included now the bp of the amplicon size in Supplementary Table 1.
- Line 243: Why are the authors' names missing?
Response: The authors' names have been duly included, addressing the earlier omission.
These adjustments contribute to the document's overall quality and ensure it meets the expected standards. Thank you for your thoughtful suggestions.
Reviewer 4 Report
Comments and Suggestions for Authors
The brief report, “The response regulator OmpR negatively controls the expression of genes implicated in the production of tilimycin and tilivalline cytotoxins in Klebsiella oxytoca,” investigates the role of OmpR in the EnvZ/OmpR two-component osmoregulation system. It shows that OmpR is a crucial negative transcription regulator for the aroX and NRPS operons, controlling the production of tilimycin and tilivalline cytotoxins in the toxigenic strain Klebsiella oxytoca.
Minor comments:
1. The manuscript needs to improve its English grammar.
2. The source of the toxigenic strain Klebsiella oxytoca MIT 09-7231 should be added to the “Materials and Methods” section.
3. What methods were used to control the concentration of Caco-2 cells? (.....into 90 μL of Caco-2 cells, totaling 1 x 104 cells.......), section “Materials and Methods”.
4. It would be interesting and practical to compare the cytotoxic results of an incubation period of Caco-2 cells of 24 hours. Section “Materials and Methods”.
5. In section “Materials and Methods”, it is advisable to provide a link to "…..guidelines provided by the manufacturer."
6. The caption to Fig. 2 mentions "lysis buffer," which needs to be characterized (composition, manufacturer) and added to section “Materials and Methods”.
The work as a whole looks like a completed stage of research and leaves a positive impression.

Author Response
The brief report, “The response regulator OmpR negatively controls the expression of genes implicated in the production of tilimycin and tilivalline cytotoxins in Klebsiella oxytoca,” investigates the role of OmpR in the EnvZ/OmpR two-component osmoregulation system. It shows that OmpR is a crucial negative transcription regulator for the aroX and NRPS operons, controlling the production of tilimycin and tilivalline cytotoxins in the toxigenic strain Klebsiella oxytoca.
We sincerely appreciate you taking the time to review our manuscript. Your insightful observations have greatly enriched our work. We have highlighted the changes made in response to your feedback in gray throughout the manuscript. Thank you for contributing to the improvement of our paper.
Minor comments:
1. The manuscript needs to improve its English grammar.
Response: We have taken proactive steps by revising all the manuscript with the advanced Grammarly Pro application. Furthermore, a native English speaker with expertise in science has thoroughly reviewed the manuscript to ensure the highest quality.
2. The source of the toxigenic strain Klebsiella oxytoca MIT 09-7231 should be added to the “Materials and Methods” section.
Response: We have included reference (9), which provides details about the origin of the strain used (lines 75 and 76).
3. What methods were used to control the concentration of Caco-2 cells? (.....into 90 μL of Caco-2 cells, totaling 1 x 104 cells.......), section “Materials and Methods”.
Response: We meticulously counted cells to ensure an accurate inoculum in each well, utilizing the trypan blue exclusion method in a Neubauer chamber for reliable results (lines 138-140).
4. It would be interesting and practical to compare the cytotoxic results of an incubation period of Caco-2 cells of 24 hours. Section “Materials and Methods”.
Response: Thank you for your valuable suggestion; we truly appreciate it! It’s crucial to emphasize that the cytotoxic effects of K. oxytoca on epithelial cells become notably evident around the 48-hour mark. This timeframe enables us to detect significant differences more effectively. We take your observation seriously and have incorporated this insight along with the relevant references in lines 147 and 148.
5. In section “Materials and Methods”, it is advisable to provide a link to "…..guidelines provided by the manufacturer."
Response: We have provided the link to the manufacturer's guidelines as indicated. Thank you for your observation, which is noted and appreciated (lines 134-136).
6. The caption to Fig. 2 mentions "lysis buffer," which needs to be characterized (composition, manufacturer) and added to section “Materials and Methods”.
Response: We appreciate your feedback on this matter. However, the manufacturer (Invitrogen) does not disclose the lysis buffer's composition, preventing us from adding this information to the "Materials and Methods" section. We hope this clarifies our position.
The work as a whole looks like a completed stage of research and leaves a positive impression.
Response: We sincerely appreciate your valuable feedback, as it has significantly enhanced the quality of our manuscript.
Round 2
Reviewer 1 Report
Comments and Suggestions for Authors
This paper has been well-edited and can be published.